# Language Guided Interpretable Image Recognition by Manifold Alignment

## Abstract

Most works of interpretable neural networks strive for learning the semantics concepts merely from single modal information such as images. However, humans usually learn semantic concepts from multiple modalities and the semantics is encoded by the brain from fused multi-modal information. Inspired by cognitive science and vision-language learning, we propose a two-stream model for learning visual semantic concepts under the guidance of natural language, where a CNN-based vision stream encodes the input image and a Bert-based language stream encodes corresponding text description. Therefore, visual and natural language features reside on different but semantically highly correlated manifolds, i.e. follow a multi-manifold distribution. We transform the multi-manifold distribution alignment problem into updating the projection matrices by Cayley transform on the Stiefel manifold and better joint representations are obtained by fusing the semantically similar features from the aligned manifold. In addition, we propose a Manifold Alignment based Prototypical Part Network (MA-ProtoPNet) to learn the semantics concepts from the joint representations, and these concepts can capture more semantic information from multi-modality. We verified the effectiveness of the manifold alignment method through experiments and the proposed framework can provide better interpretability and classification results.

## 1 Introduction

Deep learning has demonstrated remarkable performance and has been extensively utilized in various fields, such as image recognition (He et al., 2016; Huang et al., 2017) and object detection (Girshick, 2015; Ren et al., 2015). However, despite its impressive performance, deep neural networks are still perceived as a black-box model that lacks interpretability. This limitation restricts their application in high-stakes areas such as finance, self-driving, and disease diagnosis. Therefore, interpretability is crucial in these critical domains, and it is essential to understand precisely how the model makes decisions.

Interpretability of neural networks has recently gained significant attention, and self-explaining convolutional neural networks (CNNs) based on prototype learning (Chen et al., 2019a; Wang et al., 2021; Nauta et al., 2021b; Donnelly et al., 2022) have emerged as a major research direction for interpretable computer vision. Rosch (Rosch, 1973; 1975), the main representative of prototype theory in cognitive science, confirmed through a series of psychological experiments that semantic concepts consist of two factors: the prototype or best instance, and the degree of category membership, which depends on the resemblance to the prototype. Inspired by the prototype theory, the prototypical part network (ProtoPNet (Chen et al., 2019a)) and its extension works (Wang et al., 2021; Nauta et al., 2021b) first learn the set of prototypes (semantic concepts) from the images in the training set and then make predictions by comparing the similarity between the prototypes and the parts of images. This approach generates an explanation of the form "the bird is *an European Goldfinch* because the parts of the image look like the prototype parts of *the European Goldfinch*". However, most interpretable neural networks still focus on learning semantic concepts from a single modality such as images, while abundant psychological evidence suggests that humans learn more semantic information by combining language and its meaning in the physical world (Pulvermller, 2013; Günther et al., 2020).

In this study, we propose a two-stream model inspired by prototype theory and vision-language learning to jointly learn semantic concepts from visual and linguistic modalities. The two-stream model includes a CNN-based vision stream and a Bert-based language stream that encode fine-grained features from images and corresponding text descriptions. A major challenge is that features

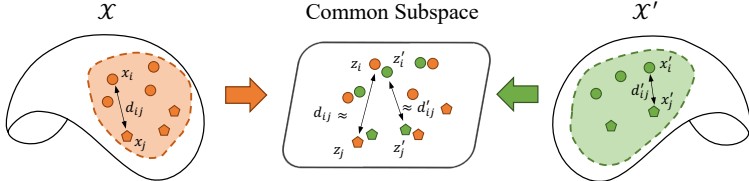

Figure 1: The basic idea of Manifold alignment: corresponding data points are mapped from two different manifolds to similar locations in a common subspace while preserving the local geometry of each manifold.

from the vision and language streams reside on different, yet semantically highly correlated, manifolds, i.e. multi-manifold distribution. To align their underlying manifolds and provide knowledge transfer across domains, manifold alignment methods (Wang & Mahadevan, 2009b; Cui et al., 2014; Pei et al., 2011; Huo et al., 2021) are designed to establish relationships between disparate datasets. Figure 1 illustrates the basic idea of manifold alignment, which aims to construct or strengthen relationships between different datasets and project data points to a common subspace while preserving local geometry structures.

To align the visual and language features, we adopt the manifold alignment method by learning two projection matrices that project them into a common subspace. In this subspace, we enforce visual and language features that share similar semantic meanings to be close, i.e. feature matching, while preserving their neighborhood relationships in the original feature spaces, i.e. local geometry preserving. We convert the manifold alignment problem into an optimization problem with orthogonality constraints on the projection matrix, taking into account the criteria of feature matching and geometry preserving. We then transform the optimization problem with Cayley transform and update the two projection matrices using a curvilinear search on the Stiefel manifold. Through the feature transformation of the projection matrix, semantically aligned visual and natural language representations exhibit similar patterns, and we fuse them in a common subspace. Finally, we propose a Manifold Alignment based Prototypical Part Network (MA-ProtoPNet) to learn semantic concepts from the fused representation and design an alternating optimization algorithm to optimize the two objective functions in manifold alignment and concept learning.

We have conducted extensive experiments on two case studies, bird species identification and flower species identification, to evaluate the performance of our model in terms of both interpretability and accuracy. To measure interpretability, we compared the output region of semantic concepts with annotated object parts, which provides a quantifiable metric that was previously missing from ProtoPNet-based methods. Furthermore, we designed a baseline model that learns concepts directly from features without manifold alignment to demonstrate the effectiveness of our proposed framework. Our model outperformed classical interpretable models in both accuracy and interpretability.

The contributions of our work are summarized as follows: (1) A new framework of interpretable image recognition guided by language is first proposed that can learn semantics concepts from both visual and linguistic modality. (2) We design a manifold alignment method to optimize the projection matrices on the Stiefel manifold by the Cayley transform, which transforms the features of multi-manifold distributions into a common subspace. (3) An alternating optimization algorithm is proposed to optimize objectives of manifold alignment and concept learning. (4) We apply quantitative metrics of interpretability for the ProtoPNet-based methods. Extensive experiments have demonstrated that reasonable utilization of natural language guidance can improve the accuracy and interpretability of the self-explaining model.

## 2  RELATED WORK

Our work is related to interpretability in image recognition, vision-language learning, and manifold alignment. In this section, we will give a brief overview of related works.

**Interpretability in Image Recognition.** Previous methods of interpreting deep neural networks can be broadly classified into two types, one devoted to visualizing the underlying patterns of the black-box model, and the other focusing on modeling clear semantic representations. There are a lot of visualization methods to compute the relevance score at each pixel-level location of the image, including the saliency-based methods (Dabkowski & Gal, 2017; Mahendran & Vedaldi, 2016; Zeiler & Fergus, 2014; Zhou et al., 2016), perturbation-based forward propagation methods (Fong & Vedaldi, 2017; Fong et al., 2019), and backpropagation-based methods (Simonyan et al., 2013; Springenberg et al., 2014; Bach et al., 2015; Zhang et al., 2018a; Montavon et al., 2017; Sundararajan et al., 2017; Shrikumar et al., 2017). But the collection of pixels cannot build the connection

with semantics concepts of humans and lack of illustrating knowledge hidden behind all activations. Therefore, some methods aim to learn semantics concepts to construct a self-explaining model during the training process, such as Zhang et al. (2018b); Liang et al. (2020); Chen et al. (2020) focus on learning interpretable filters by restricting each filter response to a specific concept. For grasping the concepts, some researchers try to represent the semantics patterns as the prototype vectors (Chen et al., 2019a; Wang et al., 2021; Kim et al., 2021; Nauta et al., 2021b; Donnelly et al., 2022; Nauta et al., 2021a). Chen et al. (2019a) proposed a prototypical part network (ProtoPNet) to learn the prototypes and can generate a part-level attention map as an explanation by calculating the distance between an input image and learned prototypes. Further, the ProtoPNet has been extended many times (Wang et al., 2021; Nauta et al., 2021a; Kim et al., 2021; Donnelly et al., 2022) and different methods have added different properties to the prototypes. Wang et al. (2021) designed a plug-in transparent embedding space, which is spanned by disentangled basis concepts on the Grassmann manifold. Nauta et al. proposed a Neural Prototype Tree that combines prototype learning with decision trees and can explain a prediction by outlining a decision path.

However, the above methods only learn semantic concepts from visual modality and lack other modal information to guide concept learning. In contrast, our model is the first framework to learn the semantics concepts from fused multi-modal information and a manifold alignment method is proposed to align the multi-modal manifold distribution. In this way, the concepts can capture richer semantic information and provide better interpretability.

**Vision-Language Learning.** Grounding language to vision is attracting increasing attention. Traditional strategies that have been used to fuse words and corresponding visual information are perceptual norms (Silberer & Lapata, 2012), bags-of-visual-word (Bruni et al., 2014; Lazaridou et al., 2014), and learnable visual features (Lazaridou et al., 2015; Chrupała et al., 2015; Kiros et al., 2018; Ailem et al., 2018). These model used for vision-language fusion have evolved with natural language processing (NLP) models, such as Latent Dirichlet Allocation (LDA) (Bruni et al., 2014), log-bilinear model (Gupta et al., 2019), Skip-gram model (Zablocki et al., 2018; Lazaridou et al., 2015), and recurrent neural network (Chrupała et al., 2015). Recently, with the development of transformer (Devlin et al., 2018; Vaswani et al., 2017), the transformer-based models have started to be used in vision-language learning and the demonstrated powerful performance (Tan & Bansal, 2019; Lu et al., 2019; Su et al., 2019; Chen et al., 2019b).

Our model relates closely to the methods (Jia et al., 2021; Radford et al., 2021; Zhang et al., 2021) which build a two-stream model to learn vision and language representation from image-text description pairs. But, we operate on fine-grained visual and natural language features in our work compared to (Jia et al., 2021; Radford et al., 2021) and capture the semantic concepts from the fine-grained vision-language representation. In addition, compared to (Zhang et al., 2021), we consider that original vision and language features come from two semantically highly correlated but different manifolds, so we design a manifold alignment method to align the vision and language manifolds.

**Manifold Alignment.** Manifold alignment has been widely used in many fields of machine learning and data mining. The main objective of manifold alignment is to align the sets of data from different manifold distributions by matching the corresponding instances of different manifold distributions and preserving the local geometry of each manifold. Depending on whether the correspondence information is available or not, manifold alignment can be classified into semi-supervised manifold alignment (Wang & Mahadevan, 2011; Lin & Tang, 2006; Ham et al., 2005) and unsupervised manifold alignment (Wang & Mahadevan, 2009b; Cui et al., 2014; Pei et al., 2011; Huo et al., 2021). Generally, the manifold alignment problem is transformed into an optimization problem that finds the projection matrices from the original spaces to a common subspace, and the projection matrices are solved in closed form by eigenvalue decomposition like Canonical Correlation Analysis (CCA) (Hardoon et al., 2004). Different from existing methods, we develop the optimization algorithm of the projection matrices based on the Cayley transform for preserving the orthogonality constraints, which can obtain the global optimal solution (Wen & Yin, 2013).

## 3 METHOD

In this section, we present our proposed Manifold Alignment-based Prototypical Part Network (MA-ProtoPNet), whose overall architecture is depicted in Figure 2. Our work proposes a manifold alignment algorithm that projects visual and language features from their respective manifolds into a vision-language aligned manifold. Section 3.1 provides a detailed description of the manifold align-

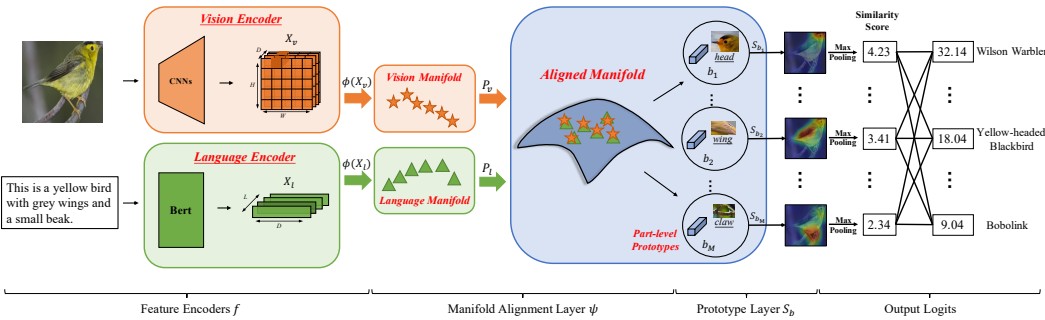

Figure 2: The overall architecture of our proposed manifold alignment based prototypical part network (MA-ProtoPNet). An input image and the corresponding text descriptions are extracted by the vision encoder and the language encoder respectively. A manifold alignment layer is designed to learn two projections for the output of encoders such that vision and language features can be aligned in a common manifold. Then the two groups of aligned features are fused by an affine transformation and the fused features are fed into the prototype layer to learn the semantic concepts. Finally, semantic concepts can be learned under the guidance of the language and the final classification is inferred from these concepts.

ment method. In Section 3.2, we propose a language-guided part-level prototype learning framework based on the manifold alignment algorithm. This framework learns part-level concepts from the semantically aligned joint representation. Finally, Section 3.3 introduces the implementation details of the two-step alternating optimization process in MA-ProtoPNet.

## 3.1 Manifold Alignment Method

We denote the input image and its corresponding text description as $\mathcal{I}_v$ and $\mathcal{I}_l$, respectively. Their features extracted by the vision encoder $f_v(\cdot)$ and the language encoder $f_l(\cdot)$ are denoted by $\mathbf{X}_v \in \mathbb{R}^{C \times W \times H}$ and $\mathbf{X}_l \in \mathbb{R}^{C \times L}$, respectively. Here, $C$ represents the number of channels of the feature maps and word embeddings, while $H$ and $W$ denote the heights and widths of the feature maps, and $L$ represents the length of the description. The vision encoder $f_v(\cdot)$ is composed of a convolutional network backbone (e.g. VGG, ResNet, and DenseNet) and additional $1 \times 1$ convolutional layers. Similarly the language encoder $f_l(\cdot)$ is based on Bert (Devlin et al., 2018) with a fully connected layer. Using extra $1 \times 1$ convolutional layers and a linear transformation, the output of vision and language encoder can be projected to the same dimensional representational space. However, without any processing, the two sets of features would be different but semantically highly correlated manifold distributions. The goal of our manifold alignment algorithm is to learn two projections $\mathbf{P}_v \in \mathbb{R}^{C \times C}$ and $\mathbf{P}_l \in \mathbb{R}^{C \times C}$ to transfer the vision and language features to a common subspace such that the two distributions can be aligned semantically. The transformed features of visual feature $\mathbf{X}_v$ is denoted as $\mathbf{Z}_v = \mathbf{P}_v^\top \mathbf{X}_v$ in the common subspace while the transformed features of language feature $\mathbf{X}_l$ is denoted as $\mathbf{Z}_l = \mathbf{P}_l^\top \mathbf{X}_l$.

The objective of the classical manifold alignment algorithm is to learn a projection that maps features from different spaces to a common subspace simultaneously matching the corresponding instances and preserving the local geometry of each manifold. Matching the corresponding instance implies that instances with comparable semantic meanings from distinct manifolds will be projected onto neighboring regions following transformation, while preserving the local geometry implies that the relationship of the neighborhood of the instances in each manifold will remain consistent after transformation. However, obtaining fine-grained annotation relationships between visual features and text descriptions is expensive and a more general manifold alignment algorithm is implemented with no available correspondence information. Therefore, solving the alignment problem without correspondence needs to find two unknown variables, i.e. the correspondence, and the projection matrix.

**Constructing the Correspondence.** The connection between different manifolds is usually constructed by some similarity measures (Cui et al., 2014; Huo et al., 2021) and the inner-product (Zhang et al., 2021) has been adopted to calculate the similarity between the visual representation and language representation, where potentially similar semantic meanings between visual features and language features have greater similarity scores. Therefore, we use the normalized inter-product $sim(x, y) = x^\top y / ||x|| ||y||$ as similarity measure and the correspondence matrix

$\mathbf{W}_{vl} \in \mathbb{R}^{(W \times H) \times L}$ of the visual features and language features is defined as follows:

$$W_{vl}^{(i,j)} = \begin{cases} 1 & \phi_i(\mathbf{X}_v) \in \mathcal{N}_k(\phi_j(\mathbf{X}_l)) \text{ or } \phi_j(\mathbf{X}_l) \in \mathcal{N}_k(\phi_i(\mathbf{X}_v)) \\ 0 & \text{otherwise} \end{cases} \tag{1}$$

where $W_{vl}^{(i,j)}$ is the element of the correspondence matrix $\mathbf{W}_{vl}$. $\phi_i(\mathbf{X}_v)$ denotes the operation of taking the $i$-th location in the visual feature map $\mathbf{X}_v$ and $\phi_j(\mathbf{X}_l)$ denotes the operation of taking the $j$-th word in the language feature $\mathbf{X}_l$. $\mathcal{N}_k(\phi_i(\mathbf{X}_v))$ and $\mathcal{N}_k(\phi_j(\mathbf{X}_l))$ represent the $k$ nearest neighbors of $i$-th visual feature and $j$-th language feature in common space. If $i$-th visual feature $\phi_i(\mathbf{X}_v)$ is in the top-$k$ neighbors of $\phi_j(\mathbf{X}_l)$, $W_{ij}^{vl}$ is set to 1 or vice versa.

**Objective Function of Manifold Alignment.** In our work, the manifold alignment is used to make visual features and language features with similar semantics close after projected to the common subspace. To achieve the above goals, the objective function can be defined as follows:

$$\min_{\mathbf{P}_v, \mathbf{P}_l} J(\mathbf{P}_v, \mathbf{P}_l) = \frac{1}{\hat{N}} \sum_{i=1}^{L} \sum_{j=1}^{W \times H} \mathbf{W}_{vl}^{(i,j)} ||\phi_i(\mathbf{Z}_l) - \phi_j(\mathbf{Z}_v)||_2^2 \tag{2}$$

where $\hat{N}$ is the number of pairs of $k$ nearest neighbors, i.e. the number of element $W_{vl}^{(i,j)}$ equals 1 in correspondence matrix $\mathbf{W}_{vl}$. If $\phi_i(\mathbf{X}_v)$ is one of the $k$ nearest neighbors of $\phi_j(\mathbf{X}_l)$ in original space, the distance between $\phi_i(\mathbf{Z}_l)$ and $\phi_j(\mathbf{Z}_v)$ will be penalized in projected space.

In the common subspace unrolled by the manifolds, the nearest neighbor relationships of each manifold are expected to be protected, i.e. the local geometry should not be destroyed. In many manifold learning algorithms (Wang & Mahadevan, 2009a; Cui et al., 2014), the local geometry is characterized by computing the local adjacency weight matrix, i.e. the similarity between instances in the manifold. According to the previous work (Anisimov et al., 2011; Shechtman & Irani, 2007), the image and sentence structure are encoded by the similarities of the visual features with different locations and language features with different words. In our work, for preserving the local geometry, the orthogonal constraints are added on the projection matrix $\mathbf{P}_v$ and $\mathbf{P}_l$, i.e. $\mathbf{P}_v^\top \mathbf{P}_v = I$ and $\mathbf{P}_l^\top \mathbf{P}_l = I$. Under the constraint of projection matrix orthogonality, the local geometry can be preserved and the proof is as follows:

$$\begin{aligned} d(\phi_{i_1}(\mathbf{Z}_v), \phi_{i_2}(\mathbf{Z}_v)) &= ||\mathbf{P}_v^\top \phi_{i_1}(\mathbf{X}_v) - \mathbf{P}_v^\top \phi_{i_2}(\mathbf{X}_v)||_2^2 \quad = \mathbf{P}_v^\top \mathbf{P}_v ||\phi_{i_1}(\mathbf{X}_v) - \phi_{i_2}(\mathbf{X}_v)||_2^2 \\ &= ||\phi_{i_1}(\mathbf{X}_v) - \phi_{i_2}(\mathbf{X}_v)||_2^2 \end{aligned} \tag{3}$$

The same proof exists for the projection matrix $\mathbf{P}_l$ and the original structure of visual features and language will be preserved with the orthogonal constraint in the projected common subspace.

For matching the semantically similar features among manifolds and protecting the local geometry of each manifold, we can combine the objective function Equation 2 with the orthogonal constraint. Firstly, we convert the Equation 2 to the following form:

$$\min_{\mathbf{P}_v, \mathbf{P}_l} J(\mathbf{P}_v, \mathbf{P}_l) = \text{tr}(\mathbf{P}_v^\top \mathbf{X}_v \mathbf{D}_v \mathbf{X}_v^\top \mathbf{P}_v + \mathbf{P}_l^\top \mathbf{X}_l \mathbf{D}_l \mathbf{X}_l^\top \mathbf{P}_l - 2\mathbf{P}_v^\top \mathbf{X}_v \mathbf{U}_{vl} \mathbf{X}_l^\top \mathbf{P}_l) \tag{4}$$

where $\mathbf{U}_{vl} = \mathbf{W}_{vl}/\hat{N}$, $\mathbf{D}_v \in \mathbb{R}^{(W \times H) \times (W \times H)}$ is a diagonal matrix with the element $\mathbf{D}_v(i, i) = \sum_{j=1}^{L} \mathbf{U}_{vl}(i, j)$. The similar with $\mathbf{D}_l \in \mathbb{R}^{L \times L}$ is a diagonal matrix with the element $\mathbf{D}_l(j, j) = \sum_{i=1}^{W \times H} \mathbf{U}_{vl}(i, j)$ and $\text{tr}(\cdot)$ is the trace.

However, considering the orthogonal constraint of projection matrices, the first and second terms are irrelevant to $\mathbf{P}_v$ and $\mathbf{P}_l$. The final manifold alignment problem can be transformed into solving the following optimization problem with constraints:

$$\min_{\mathbf{P}_v, \mathbf{P}_l} J(\mathbf{P}_v, \mathbf{P}_l) = -\text{tr}(2\mathbf{P}_v^\top \mathbf{X}_v \mathbf{U}_{vl} \mathbf{X}_l^\top \mathbf{P}_l) \quad \text{s.t. } \mathbf{P}_v^\top \mathbf{P}_v = I \ \mathbf{P}_l^\top \mathbf{P}_l = I. \tag{5}$$

Solving the above optimal solution is difficult directly since the orthogonal constraints can lead to many local optimal solutions. Therefore, we solve the optimization problem based on the Cayley transform and optimize the projection matrices on the Stiefel manifold in our work. In the Stiefel manifold, the feasible set $\mathcal{P} = \{\mathbf{P} \in \mathbb{R}^{C \times C} : \mathbf{P}^\top \mathbf{P} = I\}$ and projection matrices $\mathbf{P}_v$ and $\mathbf{P}_l$ are updated by a curvilinear search (Wen & Yin, 2013) in the feasible set through the Cayley transform:

$$\mathbf{P}_v^{(t+1)} = (I + \frac{\eta_v}{2}\mathbf{A}_v)^{-1}(I - \frac{\eta_v}{2}\mathbf{A}_v)\mathbf{P}_v^{(t)} \quad \mathbf{P}_l^{(t+1)} = (I + \frac{\eta_l}{2}\mathbf{A}_l)^{-1}(I - \frac{\eta_l}{2}\mathbf{A}_l)\mathbf{P}_l^{(t)} \tag{6}$$

where $t$ is the present step, $t+1$ is the next step, $\mathbf{A} = \mathbf{G}(\mathbf{P}^{(t)}) - \mathbf{P}^{(t)}\mathbf{G}^\top$ is a skew-symmetric matrix obtained by Cayley transform, $\mathbf{G}$ is the gradient of the objective function and $\eta$ is the learning rate. We can derive the closed-form solution for the gradient of the objective function as follows:

$$\mathbf{G}_v = -2\mathbf{X}_v\mathbf{U}_{vl}\mathbf{X}_l^\top\mathbf{P}_l \quad \mathbf{G}_l = -2\mathbf{X}_l\mathbf{U}_{vl}^\top\mathbf{X}_v^\top\mathbf{P}_v \tag{7}$$

The stochastic gradient of a mini-batch is calculated to replace $\mathbf{G}$ in each step and momentum is applied to accelerate and stabilize the stochastic gradient. Through manifold alignment, we can obtain two projection matrices to project vision and language features with similar semantics onto the aligned manifold closely.

## 3.2 MA-PROTOPNET FRAMEWORK

In this section, we will introduce the framework of manifold alignment based prototypical part network (MA-ProtoPNet) and clarify how to learn semantics concepts from cross-modal features. As shown in Figure 2, our proposed MA-ProtoPNet consists of three main parts: feature encoders including vision encoder and language encoder, a manifold alignment layer, and a prototype layer where encoders and the manifold alignment layer are introduced in the previous section. The input image $\mathcal{I}_v$ and the corresponding text description $\mathcal{I}_l$ can be transformed into $Z_v$ and $Z_l$ through encoders and the manifold alignment layer. Therefore the semantically similar representations between visual and language features can be fused as a joint representation $\mathbf{Z}_{vl}$ calculated by $\mathbf{Z}_{vl} = \alpha\mathbf{P}_v^\top\mathbf{X}_v + (1-\alpha)\mathbf{P}_l^\top\mathbf{X}_l\mathbf{W}_{vl}^\top$ in the aligned manifold, where $\alpha$ is the coefficient to balance the two streams and the shape of joint representation is $\mathbf{Z}_{vl} \in \mathbb{R}^{C\times(W\times H)}$.

In the prototype layer, we adopt the standard prototypical part network (Chen et al., 2019a) to learn $m$ basis concepts $\mathbf{B} = \{b_j\}_{j=1}^m$ with $b_j \in \mathbb{R}^C$ based on the joint representation $\mathbf{Z}_{vl}$ and every basis concept can be pre-assigned to a category, i.e. class-specific concept. Each basis concept unit $S_{b_j}$ first computes the squared $L_2$ distance $||\phi(\mathbf{Z}_{vl}) - b_j||_2^2$ between the basis concept $b_j$ and the all $1\times1$ patches of $\mathbf{Z}_{vl}$ and then convert to similarity by the function $\log((||\phi(\mathbf{Z}_{vl}) - b_j||_2^2 + 1)/(||\phi(\mathbf{Z}_{vl}) - b_j||_2^2 + \epsilon))$. If the patch $\phi(\mathbf{Z}_{vl})$ is closer to the basis concept $b_j$, the function will produce a higher similarity score. Given an input image, the similarity map with shape $W\times H$ of the $j$-basis concept can be upsampled to the input image to illustrate how strongly the basis concept $b_j$ is activated in the image and the similarity score of the basis concept can be obtained by performing a global max pooling on the similarity map. Finally, the output logits can be produced by multiplying the similarity score by the weight of the fully connected layer $h$ and a softmax function is applied to calculate the probability that the current image belongs to all categories.

We would like to learn a feature space that the patches of joint representation are clustered on at least one semantically similar basis concept of the ground truth category and separated from basis concepts of the other category. Therefore, the cluster and separation losses introduced in (Chen et al., 2019a) are formalized as:

$$\mathcal{L}_{clst} = \frac{1}{N}\sum_{i=1}^N \min_{j:b_j\in\mathbf{B}(y^i)} \min_{z_{vl}\in\phi(\mathbf{Z}_{vl})} ||z_{vl} - b_j||_2^2 \quad \mathcal{L}_{sep} = -\frac{1}{N}\sum_{i=1}^N \min_{j:b_j\notin\mathbf{B}(y^i)} \min_{z_{vl}\in\phi(\mathbf{Z}_{vl})} ||z_{vl} - b_j||_2^2 \tag{8}$$

where $N$ is the number of training set and $y^i$ is the label of instance $i$. The final optimization problem combined with the cross-entropy loss (CrsEnt) to penalize the misclassification can be formed as follows:

$$\mathcal{L}_{\text{total}} = \mathcal{L}_{\text{CrsEnt}} + \lambda_1\mathcal{L}_{clst} + \lambda_2\mathcal{L}_{sep} \tag{9}$$

where $\lambda_1$ and $\lambda_2$ is the hyper-parameter. A semantically meaningful concept space will be constructed under the constraints of these terms.

## 3.3 IMPLEMENTATION

Two optimization problems have been introduced in the above two sections. The first optimization problem is to learn the projection matrices $\mathbf{P}_v$ and $\mathbf{P}_l$ to align cross-modal manifold and the second optimization problem is the main objective to learn semantic basis concepts from the joint representation in the aligned manifold. We optimize the framework by alternating optimization that two objectives are optimized in turns. Algorithm 1 provides details of alternating optimization.
In the algorithm, $\omega_f$ denotes the parameters of the vision encoder and language encoder. $\omega_{\text{add}}$ denotes the parameters of the additional convolutional layers for the vision stream and the additional fully connected layer for the language stream. The training dataset contains the images, corresponding text descriptions, and labels $\mathcal{D} = \{\mathcal{I}_v^i, \mathcal{I}_l^i, y^i\}$. $N_{epoch}$ denotes the number of training epochs;

---

**Algorithm 1:** Alternating Optimization Algorithm

---

1  **Input:** the dataset $\mathcal{D} = \{\mathcal{I}_v^i, \mathcal{I}_l^i, y^i\}_{i=1}^n$
2  **Optimization Variables:** $\omega_f, \omega_{add}, \omega_h, \mathbf{P}_v, \mathbf{P}_l, \mathbf{B}$
3  **Parameters**: $\beta, \eta$
4  **for** $n_{epoch} = 1$ **to** $N_{epoch}$ **do**
5     **for** $t = 1$ **to** $T$ **do**
6         Sample a mini-batch $\{\mathcal{I}_v^i, \mathcal{I}_l^i, y^i\}_{i=1}^m$ from $\mathcal{D}$
7         $\omega_{add} \leftarrow \omega_{add} - \eta_{add} \nabla_{\omega_{add}} \mathcal{L}_{total}(\mathcal{I}_v^i, \mathcal{I}_l^i, y^i)$
8         $\mathbf{B} \leftarrow \mathbf{B} - \eta_B \nabla_B \mathcal{L}_{total}(\mathcal{I}_v^i, \mathcal{I}_l^i, y^i)$
9         **if** $n_{epoch} > 5$ **then**
10             $\omega_f \leftarrow \omega_f - \eta_f \nabla_{\omega_f} \mathcal{L}_{total}(\mathcal{I}_v^i, \mathcal{I}_l^i, y^i)$
11             Calculate the gradients of $\mathbf{G}$ by Eq. 7
12             Accumulate the gradients by exponential moving average $\mathbf{G}' \leftarrow \beta \mathbf{G}' + (1-\beta)\mathbf{G}$
13             **if** $t \bmod 50 = 0$ **then**
14                 Calculate learning rates $\eta_v, \eta_l$ by curvilinear search refering to (Wen & Yin, 2013)
15                 Update $\mathbf{P}_v, \mathbf{P}_l$ by Eq. 6

---

$\eta_f$, $\eta_{add}$, $\eta_B$ denotes the learning rate. For a fair comparison, after alternating optimization, we perform the same strategies named projection of prototypes and optimization of the last layer as introduced in (Chen et al., 2019a).

## 4  EXPERIMENTS

In the experiments, two case studies are conducted to evaluate our modal with other interpretable models in terms of accuracy and interpretability. The first case study is the bird species identification with 200 bird species on the CUB-200-2011 datasets (Wah et al., 2011), which is popular on the prototype-based concept learning (Chen et al., 2019a; Wang et al., 2021; Nauta et al., 2021b). The second case study is the flower species identification with 102 flower species on the Oxford Flowers datasets (Nilsback & Zisserman, 2008). Reed. et al. (Reed et al., 2016) collected the fine-grained visual description of the two datasets, and each image in CUB and Flowers have ten corresponding single-sentence visual descriptions. Some representative samples of these two datasets are shown in appendix and the visual description can accurately describe the details in the image. In addition, we adopt different CNN architectures as encoders of the vision stream and verify the generality of our model for different visual encoders.

**Evaluation Metric**: We evaluate our model in terms of accuracy and interpretability for fine-grained image recognition. For accuracy, we compare the predicted label with the ground truth category to calculate the top-1 accuracy as considered in the previous interpretable image recognition tasks. However, interpretability has not been quantified in the previous ProtoNet and the extension methods. Inspired by the previous part discovery for fine-grained recognition (Huang & Li, 2020), we designed different quantitative metrics of interpretability schemes for datasets with different annotations. In the CUB-200-2011 dataset, there are 15 part landmarks for each image and we measure the object part localization error by comparing the response region of the learned semantic concept with the annotated part landmarks. The part localization error has been adopted by (Hung et al., 2019; Huang & Li, 2020). For the datasets without part annotations such as Flower-102, we adopt the protocol of Pointing Game (Zhang et al., 2018a), which is a popular method to quantify interpretability in visualization methods (Selvaraju et al., 2017; Zhang et al., 2018a), and calculate the object localization error using the annotated segmentation. The detailed metrics are described in the supplement.

**Baseline:** One of the baselines named baseline-CNN is the original black-box model that learns only from images. To compare the effectiveness of manifold algorithms, we design another baseline model denoted ProtoPNet-VL to learn semantic concepts directly from visual features and language features in the original space without manifold alignment. The specific description of ProtoPNet-VL such as the objective function is introduced in the supplement.

### 4.1  CASE STUDY1: BIRD SPECIES IDENTIFICATION

**Dataset.** Caltech-UCSD Birds-200-2011 (Wah et al., 2011) (CUB-200-2011) is a dataset of 200 bird species for bird species recognition and contains 5,994/5,794 images for training/testing from

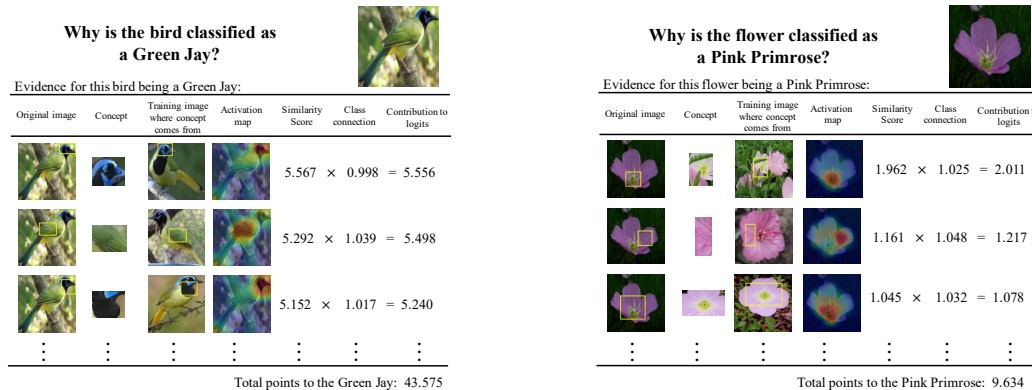

(a) The interpretable reasoning process to identify the species of a bird

(b) The interpretable reasoning process to identify the species of a flower

Figure 3: Interpretable Reasoning Process for Bird and Flower Species Identification.

200 different bird species. Each image contains a species label, 15 bird landmarks, and the bounding box of the bird.

Table 1: Top: Results of accuracy comparison on cropped bird images of CUB-200-2011 with different vision encoders; Bottom: Results of landmark localization on cropped bird images of CUB-200-2011. Normalized L2 distance (%) is reported.

| Method | VGG16 | VGG19 | ResNet34 | ResNet152 | Dense121 | Dense161 |
|---|---|---|---|---|---|---|
| Baseline-CNN | 73.3% | 74.6% | 81.2% | 80.8% | 81.8% | 82.1% |
| ProtoPNet | 77.2% | 77.6% | 78.6% | 79.2% | 79.0% | 80.8% |
| ProtoPNet-VL | 74.3% | 75.1% | 77.8% | 78.1% | 76.0% | 77.2% |
| **Ours** | **78.2%** | **79.2%** | **81.6%** | **81.7%** | **82.0%** | **83.1%** |
| Method | VGG16 | VGG19 | ResNet34 | ResNet152 | Dense121 | Dense161 |
| ProtoPNet | 21.2 | 21.0 | 21.2 | 21.3 | 20.9 | 20.7 |
| ProtoPNet-VL | 21.8 | 21.7 | 21.9 | 22.2 | 21.9 | 21.9 |
| **Ours** | **18.8** | **18.7** | **18.6** | **18.8** | **18.6** | **18.5** |

**Recognition Results (Accuracy).** We present our results of recognition accuracy with different CNN vision encoders on cropped bird images and compare them with two baselines and ProtoPNet at the top of Table 1. We also compared with other ProtoPNet-based models providing the same levels of interpretability in Table 2, where (f) means the models were trained and tested on full images and (b) means on images cropped using bounding boxes. As shown in Table 1, our model achieve higher accuracy in comparison to the ProtoPNet and ProtoPNet-VL. Learning concepts directly from raw features (ProtoPNet-VL) reduces model accuracy because of the gap between vision and natural language. Guided by natural language, the accuracy can be improved by up to 3% with the help of manifold alignment. Since each ProtPNet-based model can be understood as a "scoring sheet" as shown in Fig. 3(a), we can improve the accuracy of the model by summing the final logits values to ensemble the model, which is same with (Chen et al., 2019a). The ensemble results under the same model combination are reported and our model can improve the accuracy by 2.3% on the full images.

**Localization Results (Interpretability).** The part localization error is evaluated on various CNN vision encoders and we compare the results to ProtoPNet and ProtoPNet-VL at the bottom of Table 1. ProtoPNet-VL increases localization error and our model greatly reduces the localization error of ProtoPNet (2.4% on average). The results provide quantifiable evidence of the interpretability of our model.

**Reasoning Process.** Figure 3(a) shows the reasoning process of our model. Through concept visualization, we can find our model has learned the representative blue head, green wings, and black throat of Green Jay by training with only label supervision. Given a test image, the activation maps by calculating the distances between the patches of the image and the prototypes, and the similarity score is the maximum value on each prototype activation map. Finally, this similarity score is weighted by the fully connected weight and generates the final contribution to the category. The reasoning process makes model decisions interpretable.

**Ablation Study.** An ablation study is conducted on CUB to evaluate components of the manifold layer. We design a comparison model based on VGG-19 as a vision encoder that only uses the correspondence matrix $W_{vl}$ for feature fusion without updating the projection matrices. The model

Table 2: Comparison of our model with other ProtoPNet-based models in terms of accuracy on the CUB-200-2011 dataset.

| Level | Method | Backbone | Acc. |
|---|---|---|---|
| Part-level attention + learned concepts | ProtoShare | $1 \times$ ResNet34 | 74.7(b) |
| | ProtoPNet | | 78.6(b) |
| | **Ours** | | **81.6(b)** |
| | ProtoShare | $1 \times$ Dense121 | 74.7(b) |
| | ProtoPNet | | 79.0(b) |
| | **Ours** | | **82.0(b)** |
| | ProtoPNet | VGG19+Res34 + Dense121 | 84.8(b) |
| | **Ours** | | **85.8(b)** |
| | ProtoPNet | VGG19+ Dense 121 + Dense161 | 80.8(f) |
| | **Ours** | | **83.1(f)** |

Table 3: Top: Results of accuracy comparison on flower images of the Oxford Flowers-102 with different vision encoders; Bottom: Results of Pointing Game on flower images of the Oxford Flowers-102. Hit rate (%) is reported.

| Method | VGG16 | VGG19 | ResNet34 | ResNet152 | Dense121 | Dense161 |
|---|---|---|---|---|---|---|
| Baseline-CNN | **89.3%** | **89.7%** | **93.6%** | **94.3%** | **94.9%** | **95.1%** |
| ProtoPNet | 83.3% | 85.4% | 89.6% | 89.9% | 87.0% | 90.4% |
| ProtoPNet-VL | 82.0% | 83.3% | 88.5% | 89.9% | 90.1% | 90.2% |
| **Ours** | 83.7% | 85.7% | 90.3% | 90.3% | 90.4% | 91.2% |
| Method | VGG16 | VGG19 | ResNet34 | ResNet152 | Dense121 | Dense161 |
| ProtoPNet | 78.4 | 78.8 | 85.8 | 84.4 | 78.3 | 78.4 |
| ProtoPNet-VL | 77.4 | 78.2 | 85.5 | 85.6 | 79.9 | 79.5 |
| **Ours** | **79.6** | **79.7** | **86.6** | **87.1** | **80.3** | **80.8** |

without the updated manifold gets a higher accuracy of 79.7 % compared to the model with the updated manifold of 79.2%, but the part localization error increases to 20.2 compared to the original model's localization error of 18.7. Projection matrix update in the manifold layer slightly reduces accuracy but improves interpretability.

### 4.2 CASE STUDY2: FLOWER SPECIES IDENTIFICATION

**Dataset.** The Oxford Flowers-102 (Nilsback & Zisserman, 2008) is a dataset of 102 different categories for flower species identification. There are 10 images per class in the training and validation set (totaling 1020 images each), and the remaining 6149 images (minimum 20 per class) in the test set. Each image contains a species label and a segmentation of the flower.

**Recognition Results (Accuracy).** Our results on recognition are summarized at the top of Table 3. Since flowers do not have obvious object parts, they are more judged based on shapes and colors, thus, the ProtoPNet-based method, which is good at learning object parts, is not as good as the baseline model. Our model can improve accuracy by 0.4%-3.4% for ProtoPNet and 0.3%-2.4% for ProtoPNet-VL.

**Localization Results (Interpretability).** Further, the Pointing game results are reported at the bottom of Table 3. Our model achieves a lower localization error compared to ProtoPNet and ProtoPNet-VL. To a certain extent, our models can capture more foreground concepts.

**Reasoning Process.** Figure 3(b) shows the reasoning process to identify a flower. The model can accurately learn bright purple petals and white pistil in the center. Since flowers lack sufficient object parts and there are few images in the training set, there is some redundancy in concepts.

### 5 CONCLUSION

We introduce a novel interpretable image recognition framework that leverages natural language to learn semantic concepts. To align visual and language features that are semantically relevant but exist in different manifold distributions, we propose a manifold alignment method that optimizes the projection matrices using Cayley transform on the Stiefel manifold. We have designed an alternating optimization algorithm that addresses both concept learning and manifold alignment. Through experiments, we demonstrate that language-guided interpretable image recognition models provide better accuracy and interpretability. This is the first effort to use prototypical part networks to learn semantics from multi-modal information, and we hope that our work will inspire new approaches to semantic concept learning.

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

# A  APPENDIX

## A.1  THE BASELINE METHOD

To compare the effectiveness of manifold algorithms, we design a baseline model denoted ProtoPNet-VL to learn semantic concepts directly from visual features and language features in the original space without manifold alignment. In the supplementary, we will describe the framework of ProtoPNet-VL and clarify how to learn semantics concepts from cross-modal features. The overall architecture is shown in Figure 4. ProtoPNet-VL consists of Vision stream and Language stream and each stream can provide a decision for the classification. The prototypes is shared by the two stream and learned by the visual features and language features.

The same with MA-ProtoPNet, we denote the input image and its corresponding text description as $\mathcal{I}_v$ and $\mathcal{I}_l$, and their features extracted by the vision encoder $f_v(\cdot)$ and the language encoder $f_l(\cdot)$ as $\mathbf{Z}_v \in \mathbb{R}^{C \times W \times H}$ and $\mathbf{Z}_l \in \mathbb{R}^{C \times L}$, where $C$ is the number of channels of the feature maps and word embeddings, $H$ and $W$ are the heights and widths of the feature maps, and $L$ is the length of the description. The vision encoder $f_v(\cdot)$ consists of a backbone of the convolutional network (e.g. VGG, ResNet and DenseNet) and extra $1 \times 1$ convolutional layers. Similarly the language encoder $f_l(\cdot)$ is based on Bert Devlin et al. (2018) with a fully connected layer. Using extra $1 \times 1$ convolutional layers and a linear transformation, the output of vision and language encoder can be projected to the same dimensional representational space.

Take visual stream as an example. In the prototype layer, we adopt the standard prototypical part network Chen et al. (2019a) to learn $m$ basis concepts $\mathbf{B} = \{b_j\}_{j=1}^m$ with $b_j \in \mathbb{R}^C$ based on the visual representation $\mathbf{Z}_v$ and every basis concept can be pre-assigned to a category, class-specific concept. Each basis concept unit $S_{b_j}$ first computes the squared $L_2$ distance $||\phi(\mathbf{Z}_v) - b_j||_2^2$ between the basis concept $b_j$ and the all $1 \times 1$ patches of $\mathbf{Z}_v$ and then convert to similarity by the function $\log((||\phi(\mathbf{Z}_v) - b_j||_2^2 + 1)/(||\phi(\mathbf{Z}_v) - b_j||_2^2 + \epsilon))$. If the patch $\phi(\mathbf{Z}_v)$ is closer to the basis concept $b_j$, the function will produce a higher similarity score. Given an input image, the similarity map with shape $W \times H$ of the $j$-basis concept can be upsampled to the input image to illustrate how strongly the basis concept $b_j$ is activated in the image and the similarity score of the basis concept can be obtained by performing a global max pooling on the similarity map. Finally, the output logits can be produced by multiplying the similarity score by the weight of the fully connected layer $h$ and a softmax function is applied to calculate the probability that the current image belongs to all categories. There is a similar process in the language stream.

In the ProtoPNet-VL, we would like to learn a feature space that the patches of visual and language representations are clustered on at least one semantically similar basis concept of the ground truth category and separated from basis concepts of the other category. Therefore, the cluster and separation losses introduced in Chen et al. (2019a) are formalized as:

$$\mathcal{L}_{clst} = \frac{1}{N} \sum_{i=1}^{N} \min_{j:b_j \in \mathbf{B}^{(y^i)}} \left( \min_{z_v \in \phi(\mathbf{Z}_v)} ||z_v - b_j||_2^2 + \gamma \min_{z_l \in \phi(\mathbf{Z}_l)} ||z_l - b_j||_2^2) \right)$$

(10)

$$\mathcal{L}_{sep} = -\frac{1}{N} \sum_{i=1}^{N} \min_{j:b_j \notin \mathbf{B}^{(y^i)}} \left( \min_{z_v \in \phi(\mathbf{Z}_v)} ||z_v - b_j||_2^2 + \gamma \min_{z_l \in \phi(\mathbf{Z}_l)} ||z_l - b_j||_2^2 \right)$$

where $N$ is the number of training set, $\gamma$ is a coefficient to balance the two stream, and $y^i$ is the label of instance $i$. The final optimization problem combined with the cross-entropy loss (CrsEnt) to penalize the misclassification can be formed as follows:

$$\mathcal{L}_{\text{total}} = \mathcal{L}_{\text{CrsEnt}} + \lambda_1 \mathcal{L}_{clst} + \lambda_2 \mathcal{L}_{sep}$$

(11)

The above loss function (11) is designed to train a feature space that optimizes only the parameters of the backbone and prototype layers. For a fair comparison, we perform the same strategies named projection of prototypes and optimization of the last layer as introduced in Chen et al. (2019a).

## A.2  EVALUATION METRIC OF INTERPRETABILITY

We evaluate our model by interpretability for fine-grained image recognition. It is worth mentioning that interpretability has not been quantified in the previous ProtoNet and the extension methods. Inspired by the previous part discovery for fine-grained recognition Huang & Li (2020), we designed

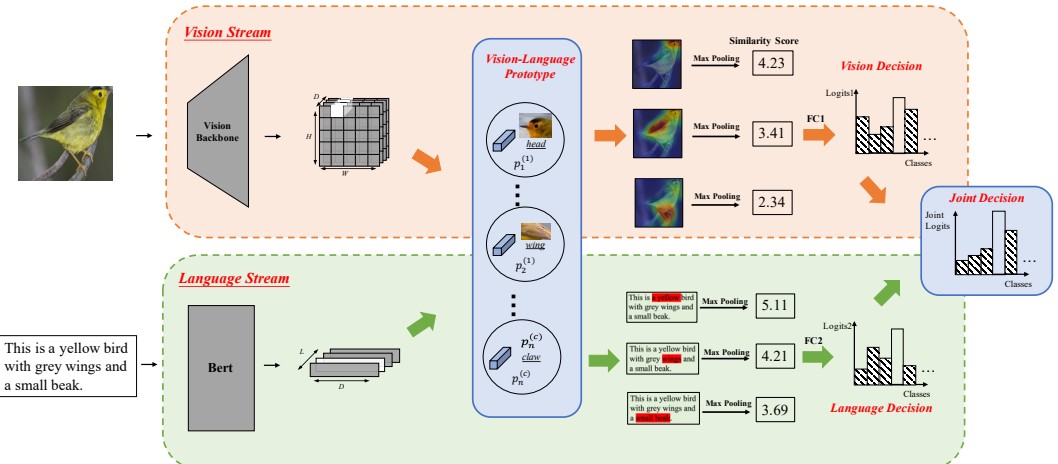

Figure 4: The overall architecture of our proposed baseline model (ProtoPNet-VL). ProtoPNet-VL consists of Vision stream and Language stream and each stream can provide a decision for the classification. An input image and the corresponding text descriptions are extracted by the vision encoder and the language encoder respectively. The vision and language features are fed into the prototype layer to learn the semantic concepts and each basis concept can generate heat maps for the inout image and text description. The similarity score for each concept is obtained by a maximum pooling operation and fed through a weighting of the fully connected layers to obtain the final classification logtis.

different quantitative metrics of interpretability schemes for datasets with different annotations. In the CUB-200-2011 dataset, there are 15 part landmarks for each image and we measure the object part localization error by comparing the response region of the learned semantic concept with the annotated part landmarks. The part localization error has been adopted by Hung et al. (2019); Huang & Li (2020). For the datasets without part annotations such as Flower-102, we adopt the protocol of Pointing Game Zhang et al. (2018a), which is a popular method to quantify interpretability in visualization methods Selvaraju et al. (2017); Zhang et al. (2018a), and calculate the object localization error using the annotated segmentation. The detailed metrics are described in the supplement.

Specifically for part localization error in the ProtoPNet-based model, we first convert the heatmap of the semantic concepts of the ground-truth class to a set of landmark locations by learning a linear regression model from training set, which is similar strategy in Hung et al. (2019); Huang & Li (2020). This linear regression model can establish the mapping from the 2D geometric centers of the concepts' heatmaps to the 2D object part landmarks in the image. The part localization errors are calculated by comparing the L2 distances between the predicted landmarks and the ground-truth part landmarks in the testing sets. The smaller the part localization errors, the more accurate the model discovers the part-level concepts. As for the pointing game in the Flower-102 dataset, we calculate the hit rate by counting the peak region of the concept's heatmaps inside the annotation segmentation. Since the semantic concepts in the ProtoPNet-based model are learned from both the foreground and background, the higher hit rate can only indicate that the model has learned less background concepts.

## A.3 ADDITIONAL EXPERIMENTS

**Implementation details of MA-ProtoPNet.** First, we resize input images into $224 \times 224$ and adopt the offline data augmentation using random rotation, skew, shear, distortion, and left-right flip to enlarge the tranining set. We adopt the Adam optimizer with learning rate $3e-3$ for add on layers and basis concepts, and $1e-4$ for the vision and language encoder. The weight decay is $1e-3$. The coefficients of $\lambda_1$, $\lambda_2$ are set to 0.8 and -0.08 respectively.

For the hyper-parameter of manifold alignment, we set the nearest neighbors $k$ as 5, and fusion ratio $\alpha$ of visual features and language features is 0.5. In the alternating optimization algorithm, the scale of exponential moving average is 0.9.

**Implementation details of ProtoPNet-VL.** The data set and optimizer settings are the same as the above method. We set the coefficient as $0.5$ and $\lambda_1$, $\lambda_2$ as $0.8$ and $-0.08$ respectively.

**Training software and platform.** We implemented our model using Pytorch and all experiments were run on 8 NVIDIA Tesla V100 GPUs.

**Our code is available in the supplementary.**

