# OpenReview forum: "Language Guided Interpretable Image Recognition via Manifold Alignment"
_ICLR.cc/2024/Conference — ICLR 2024 Conference Withdrawn Submission_

### Official Review · Reviewer_CUNM · 2023-10-30

**Soundness:** 3 good
**Presentation:** 2 fair
**Contribution:** 2 fair
**Rating:** 3
**Confidence:** 4

**Summary:**

Inspired by prototype theory, this paper proposes a two-stream model that learns semantic concepts from both visual and linguistic modalities. A manifold alignment method is applied to map the features from the two streams into a common subspace. Based on the mixed features in the subspace, a Manifold Alignment based Prototypical Part Network is proposed to achieve concept learning and interpretable image recognition. The paper conducts experiments on the CUB-200-2011 and Oxford Flowers, and the results show that the proposed MA-ProtoPNet outperforms the baseline model in terms of both accuracy and interpretability.

**Strengths:**

1. The mathematical foundation of manifold alignment is solid, with clear formulas and algorithms that are well supported.
2. The paper actively seeks methods to evaluate model's interpretability and designs different quantitative metrics for various types of datasets (object part localization error for CUB and Pointing Game protocol for Flower-102).
3. From the empirical experiments, the proposed MA-ProtoPNet shows performance improvements over the baseline model in terms of both accuracy and interpretability.

**Weaknesses:**

1. The paper claims that the proposed method achieves semantic concept learning; however, the method does not make efficient use of linguistic modality and is only used to generate a better hybrid feature for the prototypical part network. It lacks innovation in generating explanations that are easier for human to understand.
2. The design of manifold alignment is entirely based on the assumption that "the vision and language streams reside on different, yet semantically highly correlated manifolds." However, there has been no prior validation or clarification for why this assumption always holds true. This assumption seems to be not that intuitive.
3. In the experiments, compared methods are limited and outdated (ProtoPNet [NeurIPS 2019] and ProtoPShare [KDD 2021]), and it seems that ProtoPShare has not been cited correctly (in Table 2).
4. The ablation study is not sufficient as the paper only investigates the impact of the projection matrix on accuracy and interpretability. More detailed analysis of the manifold alignment module is expected. Additionally, the selection of important hyperparameters (such as α) is not cross-validated. This involves determining the contributions of visual and language modalities to the final hybrid features, which the paper does not explain in the main content but only mentions in the supplementary materials.

**Questions:**

1. Please clarify why the assumption that "the vision and language streams reside on different, yet semantically highly correlated manifolds" always holds true.
2. Please include more discussion on the ablation study and the impact of important hyperparameters (such as α) on the results.

---

### Official Review · Reviewer_LvbZ · 2023-10-31

**Soundness:** 2 fair
**Presentation:** 2 fair
**Contribution:** 2 fair
**Rating:** 3
**Confidence:** 3

**Summary:**

The authors present a method for joint vision-language representation in fine-grained recognition tasks. The idea is to allow for better explainability of the decision of a black-box neural network. Based on vision (for example, ResNet) and language (Bert) representations, two (linear) mappings to the common manifold are estimated. In this manifold, concept assignments can be made to image regions. This is done by applying a standard prototypical part network. The method is evaluated on two well-known benchmark datasets for fine-grained recognition.

**Strengths:**

The primary strength of the paper comes from the application of vision-language representations for fine-grained recognition. I also like the idea of opening the doors for the interpretability of results by semantically meaningful concepts. However, this aspect is not sufficiently demonstrated in the paper. Finally, assuming that linear embeddings into the joint manifold are reasonable, a mathematically rigorous derivation of the optimization criteria is given.

**Weaknesses:**

There are two major criticisms of the paper:
1. the manifold alignment method assumes that linear mapping can arrive at the expected properties of image regions and concepts in the joint manifold. For me, it is not clear why this shall be possible. The correspondence is measured by cosine-similarity and a K-NN decision for which K is not analyzed at all.
2. experimental results lack support for the effectiveness of the method. The baseline methods are far from the state-of-the-art of existing models (93%+ ob CUB200) on such data sets. Even concerning explainability, existing work is not considered; for example,
- Simon et al.: Generalized orderless pooling performs implicit salient matching.
International Conference on Computer Vision (ICCV). 2017.
- Simon et al.: The Whole Is More Than Its Parts? From Explicit to Implicit Pose Normalization.
IEEE TPAMI. 42 (3). 2020

In addition, I cannot see an evaluation of explainability that supports the claim by the authors. What are the benefits of this method compared to manually assigning concepts to different feature maps of a DNN? For the flowers data set, evaluation has been done empirically.

Finally, if I did not completely misunderstand the paper, where does the language modality come from? Is this part of the applied ProtoNet? I would have expected an analysis of the joint manifold, for example, showing some images and language/concepts that are close to each other.

**Questions:**

There exists work in the area of manifold alignment; for example,
- Zhao et al.: RLEG: Vision-Language Representation Learning with Diffusion-based Embedding Generation. ICML 2023.
. Li et al.: Learning Visually Aligned Semantic Graph for Cross-Modal Manifold Matching. ICIP 2019.
Why did you not consider such ideas? What do you expect would be the benefits of your alignment?

Why do you not achieve state-of-the-art on those benchmark datasets? Even for existing work no longer being state-of-the-art, the explainability of the results could be worth a comparison.

What is the advantage of your interpretation (Fig 3) over the alpha-pooling exemplar-based explanation by Simon et al. (see TPAMI paper from above)

---

### Official Review · Reviewer_76Fr · 2023-11-05

**Soundness:** 3 good
**Presentation:** 2 fair
**Contribution:** 3 good
**Rating:** 6
**Confidence:** 4

**Summary:**

This paper aims to leverage both visual and text modalities to learn more effective semantic concepts for interpretable image recognition. To this end, it constructs a two-stream model consisting of one visual encoding stream and one language encoding stream. In particular, it performs manifold alignment between the visual feature manifold and the language feature manifold by learning the project matrices by Cayley transform on the Stiefel manifold. As a result, it can learn the semantic concepts incorporating both the visual and language information by the proposed manifold alignment based ProtoPNet.

Overall, the paper is motivated well and organized well. The method is technically sound.

**Strengths:**

1. The paper is motivated well.
2. The method is technically sound.

**Weaknesses:**

1. I wonder is it possible to validate that incorporating the language can indeed improve the quality of the learned semantic concepts. For example, compare the learned semantic concepts both quantitatively and qualitatively between your method and the baseline using only the visual information.
2. The method learns the semantic correspondence between two modalities in an unsupervised way by simply minimizing the across-modality pairwise distance of the points  that within k nearest neighbors between each other in the common space and maximizing the points that outside the range of k nearest neighbors. However, how to guarantee the correctness of the initial k nearest cross-modality neighbors for a sample? I surmise that the model cannot rectify the wrongly identified k nearest neighbors automatically.

**Questions:**

Check the weaknesses above.